# Development of an Automated Design Tool for FEM-Based Characterization of Solid and Hollow Microneedles

**DOI:** 10.3390/mi14010133

**Published:** 2023-01-03

**Authors:** Yolanda Lechuga, Gregoire Kandel, Jose Angel Miguel, Mar Martinez

**Affiliations:** 1Group of Microelectronics Engineering, Department of Electronics Technology, Systems Engineering and Automation, Universidad de Cantabria, 39005 Santander, Spain; 2ENSEIRB-MATMECA, Bordeaux INP, CEDEX, 33402 Talence, France

**Keywords:** microneedle, transdermal drug delivery, transdermal biosensing, finite element modeling, graphical user interface, design tool

## Abstract

Microneedle design for biomedical applications, such as transdermal drug delivery, vaccination and transdermal biosensing, has lately become a rapidly growing research field. In this sense, finite element analysis has been extendedly used by microneedle designers to determine the most suitable structural parameters for their prototypes, and also to predict their mechanical response and efficiency during the insertion process. Although many proposals include computer-aided tools to build geometrical models for mechanical analysis, there is a lack of software utilities intended to automate the design process encompassing geometrical modeling, simulation setup and postprocessing of results. This work proposes a novel MATLAB-based design tool for microneedle arrays that permits personalized selection of the basic characteristics of a mechanical model. The tool automatically exports the selected options to an ANSYS batch file, including instructions to run a static and a linear buckling analysis. Later, the subsequent simulation results can be retrieved for on-screen display and potential postprocessing. In addition, this work reviews recent proposals (2018–2022) about finite element model characterization of microneedles to establish the minimum set of features that any tool intended for automating a design process should provide.

## 1. Introduction

During the last decade, use of microneedles (MN) for transdermal drug delivery (TDD), vaccination and transdermal biosensing has become a major field of research. The transdermal route for drug administration presents significant advantages: skin is the most extensive and accessible organ in the human body; it prevents gastrointestinal degradation or hepatic damage associated with certain compounds and it also avoids using traditional administration methods based on intravenous injections and thus provides non-invasiveness and improved user compliance [1,2,3,4]. However, this approach must deal with impermeability to many pharmacological solutes of the outermost layer of the skin, the stratum corneum [5]. In this sense, MN patches help to overcome this barrier as they can pierce the skin painlessly and create a channel for drug releasing into the skin layers underneath [6], as in Figure 1. On the other hand, MN devices have also been proposed for sample extraction, especially of blood and interstitial fluid (ISF), as key elements of point-of-care devices intended to monitor different disease-related biomarkers and molecules [7,8,9,10].

Microneedles can be mainly classified into five groups: solid, hollow, coated, dissolving and hydrogel-forming or swellable. From a structural point of view, the geometry of an MN can be, then, solid (to puncture skin and make it more permeable, or for direct releasing via coated or dissolvable materials) or hollow. In this last case, the MN must have an internal hole that allows fluid injection, or sampling, once inserted [7,11]. From the perspective of the manufacturing process, microneedles may also be classified into in-plane or out-of-plane categories depending on whether the altitude axis of the MN is parallel or normal to the substrate, respectively. Regarding in-plane processes, it is difficult to obtain two-dimensional arrays for the MN patch, and, therefore, they present serious limitations to reach a desirable amount of drug release or sampling fluid in certain applications [12]. Additionally, another limitation for in-plane MN is tip sharpness as they are fabricated by anisotropic etching and tip thickness is determined by the Si wafer [11,13]. On the other hand, out-of-plane designs present limitations in the MN length and penetration depth in the aspect ratios that can be achieved and, mainly, in its integration with other components of the system, i.e., the microfluidic and electronic parts [14].

When designing an MN patch, an extremely important issue is material selection since it is directly related to the manufacturing process and the structural stability of the patch during insertion. There are many available materials that can be selected for fabrication, such as silicon, stainless steel, glass, ceramic, polymers or resins. Silicon is one of the most extended options in the literature, mainly for its compatibility with MEMS-based manufacturing techniques and good mechanical properties to withstand the insertion process [3,12,13,15,16]. Nevertheless, silicon is a controversial material in terms of biocompatibility as it has been reported that Si particles remaining in the skin after breakage of needle tips may lead to Si-related granulomas [17,18,19]. A mixed approach of indirect fabrication remains using silicon as a base material for a master structure, and then this one is used for casting of another material more suitable for microneedle formation and biocompatibility [3]. In recent years, new techniques of additive manufacturing (AM) have gained importance as viable alternatives for the aforementioned expensive and time-consuming multi-step fabrication processes. In these proposals, computer-aided design tools (CAD) permit easy optimization and modification of the designs, which are later materialized via 3D printing machines [10,20,21].

Finally, finite element analysis (FEA) has been extendedly used in recent years by MN designers as a powerful and versatile tool that prevents utilizing, in the first phases of the design process, expensive and time-consuming experimental trials to determine the most suitable structural parameters for the MN [22]. Finite element modeling (FEM) software permits developing a wide range of static and dynamic simulations, where the critical loads before failure and insertion process parameters can be obtained. These analyses require suitable MN and/or skin models, which could be assisted and automated via CAD tools.

To the best of the authors’ knowledge, many proposals make use of different CAD software to facilitate geometrical description of their designs [10,20,21,23,24], but none comprises automation of the design process, from selection of modeling and geometrical parameters to analysis of the corresponding FEM simulation results with minimum user interaction.

In this work, a novel design tool for MN design based on mechanical characterization by finite element analysis is proposed. The MATLAB-based tool presents a graphical user interface (GUI) that permits personalized selection of the basic characteristics of the mechanical model of a single microneedle or an MN array, such as solid or hollow structure, conical or pyramidal geometry, height, base and tip areas, bore radius and offset, number of elements forming a one- or two-dimensional array, interspacing, element type or meshing size. Once the MN model has been configured, the tool exports all the selected options to an ANSYS batch file including not only this dataset but also all the necessary instructions to run automatically both static and linear buckling analyses for axial load reproducing skin resistance to puncture. This batch file can be executed directly, both in interactive and batch mode, under a Mechanical ANSYS APDL environment, and the subsequent simulation results are retrieved by the tool for on-screen display. This additional feature allows the designer to compare the simulation results with analytical and/or experimental data, to apply optimization algorithms or to derive mathematical expressions relating a mechanical response with a certain parameter variation, to name a few examples of potential applications.

Additionally, this paper includes a detailed review of significant microneedle design proposals that have been published in the last five years (2018–2022) where an FEM-based characterization has been developed for performance evaluation. The purpose of this review is determining the minimum set of features that any tool intended for automating an MN design process should include to satisfy the MN designers’ needs.

## 2. Materials and Methods

One of the most important phases during design of a microneedle patch consists of characterizing the mechanical properties of the MN so as to ensure its structural integrity during the insertion and extraction processes as this will maximize its efficiency for delivering drugs or sampling fluids. Normally, mechanical testing and simulation are oriented towards determining the maximal axial force that the structure can endure without failure [25]. The two main phenomena that can cause fracture in the MN are compression and buckling, or lateral deflection under axial load. This last one is considered the main contribution to calculus of a critical failure load [26] according to Euler’s formula [27]:(1)$$P_{cr}=\frac{\pi^{2}EI}{{\left(KL\right)}^{2}}$$
where *P_cr_* is the critical load (compression load on column), *E* is the Young’s modulus of the material, *I* is the moment of inertia, which depends on the geometry, *L* is the length and *K* is a factor related to the effective length of the column, whose value can be 2 or 0.699 depending on whether the column is free or pinned at its tip, respectively. As can be derived from Equation (1), critical load is directly proportional to the Young’s modulus or natural strength of the material and inversely proportional to the square of the MN length.

Further, the necessary force that must be applied to the MN to puncture the skin, called insertion force, is highly dependent on the contact area of the MN, and it must overcome the skin resistive pressure, which is considered to have an approximate value of 3.18 MPa [28].

The safety margin (SM) is defined as the ratio between the failure force (normally the critical buckling load) and the insertion force. Therefore, to ensure structural integrity for the MN during insertion, this safety margin must be greater than unity.

In order to develop an accurate and comprehensive study of the insertion process by finite element analysis (FEA), a functional skin model is mandatory. In this sense, the problem faced is extremely difficult to deal with due to the complex structure and changing mechanical properties of the skin, depending on the area considered for insertion and other biologically related causes of variability, such as gender, age, illnesses and/or environmental factors [25].

Human skin, apart from being the most extensive organ in the body, can be basically divided into three layers, i.e., epidermis (0.05–0.2 mm thick), dermis (1.5–3 mm thick) and subcutaneous tissue, with variations depending on the body area [29]. Accordingly, the MN must be designed so that it could reach the viable dermis, where drug delivery and interstitial fluid (ISF) extraction is most effective [30], but without arriving at the pain receptors located in the dermis layer [31]. Regarding the epidermis, the stratum corneum is the uppermost layer composing this, in turn, multilayer structure [32] and, thus, the main focus when modelling the skin puncture phenomenon.

There are multiple approaches found in the literature that deal with the difficult issue of modeling skin in a trade-off between accuracy and complexity. Globally, the skin tissue could be initially regarded as an isotropic viscoelastic material [33]. However, the most extended and simplest model used considers linear elastic behavior for a monolayer or multilayer 2D or 3D skin block, where each of these layers is defined with different mechanical parameters (Young’s modulus, Poisson’s ratio, yield strength and density) [34,35]. On the other hand, more advanced models consider hyperelastic mechanical response for, at least, the stratum corneum and epidermis layers (the ones that will be reached by the MN) for a cylindrical or prismatic 3D skin block [36,37,38].

Regarding the type of mechanical analysis performed for testing the MN performance and resistance, there are two main strategies that could be pointed out: the first one consists of a static analysis where an axial force, or the resistive pressure of 3.18 MPa, is applied to the tip of the MN, and a compression and/or buckling analysis is developed. The other approach is based on an insertion process simulation, where one of the aforementioned skin models is included and the axial load is applied or substituted by a displacement in such a way that the penetration speed is constant and so low that a quasi-static response can be considered. In this last case, the insertion force is measured when the stratum corneum fails (skin puncture or piercing), and, in some cases, an erosion algorithm is included to mimic the process beyond that point [36,39,40]. This methodology consists of removing those elements in the skin model that reached a certain condition, normally a failure-related one. This kind of simulation permits characterizing, among all, the penetration efficiency of a single MN or an array for optimization of the application strategy.

In view of the distinction thereof, a review of the latest proposals (2018–2022) for designing MNs with an FEM-based characterization has been included in the following subsections. It is worth noting that there is a significant number of different approaches that can be found in the literature. The purpose of this work is not including all the existing ones but taking a significant number of examples that permit determining the minimum features that a design tool intended for automating the design process of MN should have to be really useful for current and future designers. In this sense, reviews by Bodhale et al. [11] for proposals previous to 2010 and by Makvandi et al. [25] or Yan et al. [41] in the most recent years are highly recommended for further detail.

Table 1 and Table 2 summarize the most significant data of the FEA simulations developed in each proposal, mainly: type of microneedle, according to the structural classification into solid or hollow and in-plane or out-of-plane; manufacturing materials and their mechanical properties; geometries considered for the analysis; fixed or variable dimensions for the MN; type of FEA performed and applied loads; main simulation results; number of MNs included in the array; manufacturing process and intended application.

### 2.1. Proposals Based on Static Structural Simulation

Mamun et al. [42] propose a wet etching process to sharpen the wedge-shaped tip of silicon in-plane microneedles with the aim of improving insertion. The experimental and analytical results, before and after etching, demonstrate that the insertion, free bending and maximum buckling forces were all reduced, as well as the maximum bending stress. In this case, the FEM analysis, which included axial and transversal loads for structural characterization, was planned to support the calculations and experimental measurements, also explained in the document.

Villota et al. [43] chose a bevel cylinder as the geometry under study, although two different tip terminations are considered. On the one hand, the traditional 1-bevel tip is characterized by a single surface with a 45° tip angle. The other option is a 3-bevel tip with three penetration areas and two shear angles of 30 and 35°, inspired by the eponymous hypodermic needle type. Further, variations in the inner and outer diameter of the microneedle are considered for the FEM analysis. The simulation procedure consisted of fixing the minimum inner/outer diameter and varying the outer/inner one till the maximum von Mises stress with a safety factor of 4 (18.25 MPa) is reached, when an axial load of 0.5 N is applied. With the data obtained, a fourth-order polynomial regression approach was used to propose a mathematical model that generalized the relationships between the inner and outer diameters. Regarding the 3D printing prototypes, the resolution was not totally satisfactory as some clogging occurred in the inner holes for both minimal and maximal dimensions. For that reason, Villota et al. admit that further improvement in this sense is needed.
micromachines-14-00133-t001_Table 1Table 1Review of proposals with an FEM analysis based on static and/or buckling simulation.Type of MNRef.Material UsedMaterial PropertiesGeometryDimensionsApplied LoadReported ResultsArray SizeFabrication MethodApplicationSolid In-plane[42]SiliconE = 169 GPaσ_y_ = 7 GPaRectangular with sharp tip and triangularL = 1.5–3 mmW_b_ = 300–500 μmT_b_ = 500 μmθ = 5–30°F_t_ = 2.5 N with P_in_ = 6.36 MPaPost etching Stress 2.85 GPaBucklingSF > 11 × 11 × 5Post-CMOS compatibleN/rHollow Out-of-plane[43]Class I biocompatible resinE = 2.9 GPaσ_y_ = 73 MPaν = 0.351-bevel tipL = 450 μmD_b_ = 208–250 μmD_i_ = 30–134 μmθ = 45°F_a_ = 0.5 NMin D_i_/max D_b_Stress 12.4 MPaStrain 15.4 × 10^−4^1 × 13D printingTDD3-bevel tipθ_1_ = 30°θ_2_ = 35°Stress 15.9 MPaStrain 13.6 × 10^−4^Solid and Hollow Out-of-plane[35]Hollow: PLAE = 865.4 MPaσ_y_ = 22.9 MPaK = 54.80e = 3.12%Quad-groovedY-hollow single side-openL = 400 μmθ = 20 μmD_b_ = 200 µmD_inlet_ = 40–60 µmD_outlet_ = 20 µmAngle = 15°W_junction_ = 30 µmF_t_ = 0.7 NP_in_ = 3.18 MPaStress 3.4 MPa Deform. 0.81 µm1 × 1N/rSample extractionSolid Out-of-plane[44]Polycarbonate (PC)E = 2.0–2.4 GPaσ_y_ = 55–75 MPaν = 0.37–0.39d = 1.2–1.22 gcm^−3^ConeAL = 200 μmD_t_ = 30 μmD_b_ = 150 µmBL = 550 μmD_t_ = 50 μmD_b_ = 250 µmCL = 800 μmD_t_ = 100 μmD_b_ = 400 µmP_in_ = 3.18 MPa (compression and buckling) P_in_ = 3.18 MPa with angle 30° (bending)Compression and buckling:Max stress 2.8–2.9 MPaMax strain 1.2 × 10^−3^Bending:Max stress 6.8 MPa (Type B)Max strain 2.8 × 10^−3^ (Types B and C)1 × 1HotembossingTDDPolymethyl methacrylate (PMMA)E = 1.8–3.1 GPaσ_y_ = 50–77 MPaν = 0.35–0.4d =1.17–1.20 gcm^−3^PLAE = 3–4 GPaσ_y_ = 40–60 MPaν = 0.36d =1.21–1.43 gcm^−3^E: Young’s modulus, ν: Poisson’s ratio, σ_y_: yield strength, d: density, K: stiffness, e: elongation, L: length, W_b_: base width, T_b_: base thickness, θ: tip angle, D_i_: inner diameter, θ_1,2_: secondary tip angles, D_b_: base diameter, D_t_: tip diameter, F_a_: axial force, F_t_: transversal or bending force, P_in_: insertion pressure.

This innovative work [35] presented by Ahmad et al. proposes a design for MN with a dual release pattern and a bore for sample extraction to monitor continuously glucose concentration, and also to automatically provide a sustained and instant release of insulin dose in a non-invasive close-loop approach, known as “artificial pancreas”. The design is based on multiple outer grooves to reduce the insertion force by minimizing friction resistance with the tissue and on the unique use of dissolving and biodegradable materials. Two different models have been fluidic and mechanically analyzed by simulation: one for insulin delivery (solid-based and made of PVP and HPMC dissolving polymer) and another one for sample extraction (hollow and made of biodegradable PLA). More specifically, the sampling-oriented design is based on Y-hollow geometry with single-side opening that reduces vortex formation during extraction. As far as the structural analysis is concerned, axial and transverse loads have been applied to the MN tip, considering a 3.18 MPa value for skin resistance and a bending force of 0.05 N and 0.07 N for solid and hollow MN, respectively, taking the maximum material yield strength as a reference.

Abubaker and Zhang [44] analyse the mechanical behavior of solid polymer MNs when an axial pressure of 3.18 MPa is applied to the tip of a conical structure to assess the resistance against compression and buckling of three different polymers as manufacturing materials for the MN. This same pressure but applied with an angle of 30° is used for bending analysis of the same set of 9 MNs (three polymers for three different dimensions for the cone). Finally, the PMMA was chosen to fabricate the MN prototype by hot embossing process, and the manufacturing parameters were selected by applying Taguchi method.

### 2.2. Proposals Including a Simulation of the Insertion Process in the Skin

In Henriquez et al. [23], an FEM analysis is performed considering different geometries and building materials for the MN. The geometries included in the study are: conical, pyramidal, traditional and a proposed sting-bee topology. The so-called “traditional case” is, in reality, a beveled-tip cylinder. The diameter base and MN length are identical for all the topologies under study, and all of them include an eccentric channel for drug delivery with a fixed diameter of 90 μm. Regarding the materials considered for the analysis, these were silicon, stainless steel, polylactic–coglycolic acid or PLGA, polylactic acid or PLA and resin, pointing out that silicon and resin are not considered as biocompatible. Then, five buckling modes were simulated to extract the load factors that show the critical force that the MN could resist depending on the type of deformation to which it is exposed. To model the interaction between the MN and the skin, the mechanical properties of the different material (MNs and skin layers) have been considered as linear; the contact between surfaces (without separation) permits a maximum stress of 28.7 MPa (fracture stress of the outer layer of the skin); this contact is considered as friction-free, and displacements in the perpendicular axes to the skin surface are permitted. As a result, the best geometries in terms of structural stability are conical and pyramidal ones. However, those who achieve the necessary stresses in less displacement to ensure penetration of the outer layer of the skin are conical and beveled shapes. Stainless steel is established as the best option in terms of structural behavior and biocompatibility, but PLGA is selected as the second suggested option due to the manufacturing difficulty inherent to stainless steel material.

Ebrahiminejad et al. [39] propose a fabrication process for thermoplastic MNs from cyclo-olefin polymers (COP) using hot embossing on polydimethylsiloxane (PDMS) soft molds. The experimental characterization considered bending, buckling and tip blunting effects by developing compression tests. Further, an in vitro insertion test on porcine back skin was developed considering different application methods and a skin stretcher mechanism that was designed to mimic skin in vivo behavior. In this proposal, the FEA focused on simulation of the insertion process of a single MN into a three-layered skin model characterized by use of an erosion algorithm to delete those elements that had reached their failure stress, as well as a sphere of influence technique for the MN tip to obtain more accurate results in the piercing zone. The results presented a 0.18 N peak force before the typical sudden drop when piercing is reached and a safety margin of 7.16, this last one defined as the ratio between the buckling failure force and the simulated insertion force.

In this rather complete work [40], Xenikakis et al. propose a hollow MN for transdermal peptide delivery. A 3D printing technology combined with LCD vat polymerization process is utilized for fabrication of the MN. Different geometries (cone, square pyramid, screw or tapered-cone and triangular pyramid) and varying parameters, such as better printing angle, are analyzed with the objective of optimizing the hollow and sharp tip formation. Finally, a curved triangular pyramid topology was chosen as the one to provide better printing repeatability and constant trend to sharper tips for this particular fabrication method. Focusing only on the mechanical analysis presented in this paper, experimental penetration and compression tests were developed using human skin and Strat-M^®^ membranes, equivalent to the three skin layers. In parallel, dynamic finite element analysis was performed to replicate and compare the insertion process on both the human skin and the Strat-M^®^ membranes. In these simulations, a previously reported model for the three-layer skin block was used [45], as well as an erosion algorithm to mimic the failure and separation process of the skin elements. As the MN length is not enough to penetrate the hypodermis layer, this one was modelled as an elastic material, contrary to the hyperelastic failure mechanism considered for the other two layers. Simulation and experimental results showed that skin piercing occurred at similar forces for both materials, and, thus, they also proved the potential of this synthetic membrane as a skin alternative for mechanical-based research. Additionally, axial force simulation was conducted on the triangular and curved MN array to obtain the yield and failure strength in both cases, and, therefore, to conclude that the measured values were appreciably above the previously obtained insertion force.
micromachines-14-00133-t002_Table 2Table 2Review of proposals with an FEM analysis based on the simulation of a skin insertion process.Type of MNRef.Material UsedMaterial PropertiesGeometryDimensionsReported ResultsArray SizeFabrication MethodApplicationHollowOut-of-plane[23]SiliconE = 150 GPaσ_y_ = 165 MPaν = 0.28d = 2.33 gcm^−3^ConicalPyramidalBeveledSting typeL = 600 μmD_b_ = 300 μmD_i_ = 90 μm (eccentric)For stainless steelconical MNDisplacement for skin penetration 0.17 mmMax application force 0.16 NMax lateral force 0.0516N1 × 1N/rTDDStainlesssteelE = 193 GPaσ_y_ = 207 MPaν = 0.31d = 7.75 gcm^−3^Polylacticcoglycolic acid(PLGA)E = 2.7 GPaσ_y_ = 28.22 MPaν = 0.25d = 1.11 gcm^−3^Polylactic acid (PLA)E = 1.28 GPaσ_y_ = 53.45 MPaν = 0.36d = 1.25 gcm^−3^ResinE = 2.2 GPaσ_y_ = 33.7 MPaν = 0.26d = 1.16 gcm^−3^SolidOut-of-plane[39]Cycloolefin polymers(COP)E = 2.1 GPaν = 0.48d = 1.01 gcm^−3^Tapered-coneL = 1050 μmD_b_ = 245 μmF_in_ 0.18 NSM 7.161 × 1Hot embossing on PDMS soft moldsTDDHollowOut-of-plane[40]Class IIa biocompatible resin (MN) PLA (reservoir)E = 1.4 GPaσ_y_ = 135 MPaTriangular and triangular-curved pyramidL = 900 μmD_t_ = 50 μmS_c-t-c_ = 3000 µmF_in_ 0.56 N (skin) 0.57N (membrane)Stress 20.08 MPa Compression test:Yield strength21 N/MNFailure strength42 N/MN3 × 33D printing with LCD vat polymerizationTDDSolidAndHollowOut-of-plane[35]Solid:Polyvinyl-pyrrolidone (PVP) and Hydroxy-propyl methyl-cellulose (HPMC)E = 865.4 MPaσ_y_ = 22.9 MPaK = 54.80e = 3.12%ConeTri-grooved Quad-grooved Penta-grooved Hexa-groovedL = 400 μmD_t_ = 20 μmD_b_ = 200 µmStress 3.4 MPa Deform. 3.39 µm1 × 1N/rTDDSolidOut-of-plane[38]StainlesssteelN/rConeL = 625 μmand 2100 μmD_t_ = 36 μmD_b_ = 180 μmand 600 µmS_c-t-c_ = 156–1000µm10% strain (1 MN):F_in_ reduced 16%PE increased from 84 to 99%Limit for MN interspacing 0.5 mm5 × 5 1 × 5 1 × 13D printingN/rSolidOut-of-plane[46]PLAE = 3.5 GPaσ_y_ = 47 MPaν = 0.36d = 1.25 gcm^−3^ConeTapered-cone Pyramid Beveled-tipL = 400 μmD_b_ = 100–250 µmSide = 100–200 µmS_c-t-c_ = 1000 µmD_b_ = 100 µm:all failedD_b_ = 200 µm:all penetratedD_b_ = 150 µm:only tapered-coneLargest volume and minimum insert force for the tapered-coneN/rMEMS technology (master) and Micro hot embossingTDDSolid(simulation)Hollow(fabricated)Out-of-plane[47]Acrylic photo-curable resinE = 2 GPaCylinder (simplified version of the hollow fabricated device)L = 1000 μmD_b_ = 100 µmMaximum puncture resistance reduced by 20% with the sheet1 × 13D laser lithographySample extraction (blood)HollowOut-of-plane[48]Silicon core and Hafnium oxide HfO_2_ outer layerE = 74.2 GPaν = 0.253Tubular with square cross-sectionL = 50 μmW_b_ = 10–50 μmT_w_ = 0.2–0.9 μmS_c-t-c_ = 20–80 µmThe shorter microneedle spacing, the smaller maximal height difference of skin surface and thesmaller maximal stress in skin3 × 3Deep reactive ion etching of silicon and atomic layer deposition of HfO_2_TDDE: Young’s modulus, ν: Poisson’s ratio, σ_y_: yield strength, d: density, K: stiffness, e: elongation, L: length, D_i_: inner diameter, D_b_: base diameter, D_t_: tip diameter, S_c-t-c_: center to center spacing, W_b_: base width, T_w_: wall thickness, F_in_: insertion force, PE: penetration efficiency, SM: safety margin.

Ahmad et al. [35] also performed a mechanical dynamic analysis on a three-layer skin model to obtain the minimum insertion force for tri-, quad-, penta- and hexa-grooved models. Simulation results showed that a tri-grooved pattern minimized contact area and, thus, insertion force but reduced the MN volume, making it unsuitable for hollow MNs and sample extraction. In this case, the quad-grooved geometry provided better overall performance.

Shu et al. [38] claim that skin tension has proved its influence on microneedle insertion with experimental evidence, but it is omitted from any finite element model. Additionally, in many cases, a single MN instead of an array including a backing plate, is normally simulated, neglecting the overall influence of the complete system on the insertion mechanics. For those reasons, this work proposes a finite element model where a solid stainless steel array is simulated considering varying inter-spacing, two different lengths for the MNs as well as separate single MNs with and without backplating for comparison. On the other hand, the dynamic simulation performed to characterize the insertion process considers a multilayer 3D hyperelastic, anisotropic pre-stressed model for the skin. Further, the failure criterion for the skin is based on von Mises stress and an element deletion algorithm. Simulation results confirm the importance of including skin pretension in the model as the penetration force was reduced inducing strain, and this also shortened the penetration time. Additionally, increasing the pretension values also led to a reduction in local deformation (mouth angle) and rising penetration depth. On the other hand, MNs spacing proved to have a significant impact on the patch behavior below a certain limit according to “the bed of nails” effect [49]. Finally, the rigid flat backing plate showed how much it could impair the penetration efficiency when considered in the model. A controversial issue about this paper is mainly the statement that skin pretension reduces insertion forces and improves penetration efficiency. Even Shu et al. themselves admit that other published studies refute this point [50,51,52]. They justify this apparent contradiction, affirming that it could be partially explained by taking into account the differences in the boundary conditions on the skin, as well as the skin area selected for testing.

Chang et al.’s work [46] proposes an MN array of PLA fabricated by combination of an MEMS process to obtain the master array, a mold made of polydimethylsiloxane (PDMS) and a micro-hot-embossing manufacturing approach, to finally obtain the polymeric MN array. Different geometries were tested by finite element simulation when the insertion process is considered to select the most suitable topology for a given MN length and different base dimensions. Finally, a prototype was fabricated and tested on human skin in a puncture experiment that consisted of establishing transepidermal water loss after applying the MN array.

In an attempt to mimic nature solutions for human problems, Yamamoto et al.’s work [47] proposes a puncturing mechanism that tries to minimize skin deformation and MN buckling failure by adopting an MN geometry based on the mosquito’s proboscis. The mosquito’s mechanism for biting is based on a seven-part proboscis where the labium supports the rest of the structure while adheres to the skin to prevent the complete system from buckling and the skin from deforming. The proposed MN solution uses a jig-integrated needle fabricated with 3D laser lithography as puncturing device, combined with a mechanism that emulates the supporting effect of the mosquito’s labium. This mechanism consists of a polyethylene terephthalate (PET) sheet adhered to the skin that includes a hole at the insertion point whose diameter is slightly higher than that of the MN. This solution has been tested by FEA using a PDMS artificial skin layer (viscoelastic) with a 0.1 mm width PET layer (elastic) on top and a fully adhered condition between them. To prevent the puncture object from rising, an air layer has been also added. The fabricated design includes a buckling prevention jig that acts as a holder for the MN and a skirt-like base for increasing contact area. Contrary to what could be expected from the elaborated design that was fabricated, the FEA model is surprisingly simple: a solid unsharpened cylinder puncturing artificial skin instead of a model closer to the final design of MN (at least with a sharp tip) and the typical 3-layer model for human skin.

The work by Zhang et al. [48] has two main proposals: On the one hand, a fabrication method for hollow MNs based on a deep reactive ion etching procedure to create a silicon core of tubular structure with square cross-section, and later an atomic layer deposition technology (ALD) to generate a hafnium oxide (HfO_2_) film. On the other hand, an FEA where a 3 × 3 array of MNs of the aforementioned material has been simulated for different interspacing in order to evaluate the skin piercing result when this separation is decreased according to “the bed of nails” effect. In these analyses the symmetry of the MN was considered and only one quarter of the tubular structure was built in the model.

### 2.3. ANSYS Mechanical APDL

ANSYS Parametric Design Language (APDL) provides the user with a wide range of flexible and powerful features. It permits defining as a parameter any part of the model, such as geometry, loads or material properties. This possibility facilitates design variations intended for optimization, root cause analysis or detailed exploration of alternatives as the process will simply consist of changing one or several parameters and rerunning the simulation. Further, ANSYS APDL allows creation and execution of macros, handling arrays of parameters and includes basic logic instructions (do, repeat, if, else, then…).

In addition, the ANSYS Mechanical APDL software family provides other interesting features apart from APDL coding, such as optimization (requires a parameterized input file), probabilistic (random variation of input parameter), submodeling (two-step process with coarse and fine meshing to obtain the boundary conditions and the final results, respectively), substructuring (a stiffness matrix replaces a section of the model created separately as a substructure), user-programmable features (elements, loading, materials and commands) [53].

In order to communicate with ANSYS, the user can select among three options: the graphical user interface (Mechanical APDL user interface M-APDL-UI), the command prompt window located within the M-APDL-UI and input and batch files.

The M-APDL-UI is undoubtedly the easiest and most friendly way to interact with the software but presents important limitations. Following the necessary steps to define a model using the GUI is slower than making use of the commands directly. Additionally, the GUI offers fewer of capabilities than other options do.

For those users that have better knowledge and longer experience using ANSYS, the command prompt permits introducing commands directly while still benefiting from the GUI interactivity instead of using the menus, in a faster and more efficient way.

Finally, the input and batch text files follow the structure of one command per line, and the main difference between them is that the second must start with a first line of code that includes a /BATCH command. This instruction permits the code to be executed in the batch mode, without interactivity or real-time feedback, while running in the foreground. However, both input and batch files can be called in the interactive mode by using the Utility menu in the GUI or partially or totally pasted into the command prompt window. In this case, ANSYS will provide feedback messages and graphic content while running the file [54].

### 2.4. MATLAB App Designer

Released for the first time with the R2016a MATLAB version, App Designer is an interactive environment intended for rapid and intuitive development of app layouts and programming [55]. An app is a program that includes a graphical user interface so that the person utilizing it could provide input data and obtain output data in return in a more visual and user-friendly way than a text-based environment.

App Designer offers an extensive library of interactive UI components, as well as a grid layout manager to distribute and arrange the different components on screen. It also provides useful features as callback functions, which are activated by a certain event or interaction with the user, or the possibility of combining multiple apps in a single environment and interchange data among them.

This environment has been used in this work for development of a graphical user interface or GUI-based design tool of microneedles for both drug delivery and sample extraction applications. The tool provides a batch file with a mechanical model that can be directly simulated in ANSYS Mechanical APDL software, and, besides that, this app has been programmed to retrieve output simulation data to display some significant results on screen that can be later processed or compared as desired.

## 3. Results

Taking into account what has been elaborated in the previous section, it can be concluded that any microneedle design process should include, at least in the early stages, a structural finite element analysis to characterize its mechanical resistance in order to ensure that no failure is going to occur during the insertion process. Further, this mechanical performance needs to be carefully analyzed to accurately predict the penetration efficiency of the MN patch to determine if the obtained value is suitable for the particular drug delivery or sample extraction application it has been intended for.

With the aim of facilitating and automating the MN design process based on FEM simulation, a MATLAB tool has been developed following the workflow depicted in Figure 2.

The first step consists of defining the structure for the MN. There are important parameters here that influence the mechanical and fluidic response of the MN design and thus must be considered, such as the manufacturing material, the geometry of the puncturing device, the corresponding dimensions, if the structure is solid or hollow and, in this last case, if the inner hole is eccentric or not. Second, the selected topology, once properly defined, has to be modeled by finite elements that the FEM software could handle. In other words, the CAD model resulting from the previous step has to be meshed in a trade-off between computational cost and accuracy.

Once the MN structure has been geometrically defined and correctly modeled, it is ready for FEM simulation. In our case, we have selected the aforementioned ANSYS Mechanical APDL environment to develop a basic static analysis where the insertion pressure of 3.18 MPa is fixed as the main axial load. After retrieving the resulting stress data from this postprocessor module of the simulation, another structural analysis is configured for a subsequent simulation step: a linear buckling analysis. When ANSYS has finished both, the von Mises stresses and top displacements are retrieved and stored to display them on an appropriate panel of the designed MATLAB tool.

### 3.1. MATLAB Graphical User Interface (GUI)

This MN design tool has been developed with the objective of minimizing the effort required not only to build an accurate-enough model of the structure for further FEM characterization but also to evaluate the importance of geometrical parameters on the mechanical strength of microneedles.

In this subsection, the MATLAB-based tool is described, focusing on its features and graphical interface. An example of conical hollow MN has been chosen to better illustrate practical use of the different options provided by the tool.

Figure 3 shows the main screen of the graphical interface of the program, divided in two sections. On the left, the configuration panel can be easily distinguished, with all its menus for definition of the model and generation of the batch file, ready for simulation. On the right side is the results panel that will graphically display the main results obtained from the FEM simulation in ANSYS.

#### 3.1.1. Configuration Panel

This panel is intended to offer the possibility of manual selection of the topological and modeling options of the design. Figure 4 presents an enlargement of the upper part of this left panel, where the shape and element type for the model can be selected in two corresponding drop-down menus.

Currently, the tool offers four different geometries for the MN design, which are the cone, the triangular-based pyramid, the square pyramid and the curved cone (Figure 4a). On the other hand, there are three available element types for definition of the mechanical model (Figure 4b).

SOLID185, which is an 8-node structural solid;SOLID186, a 20-node structural solid;SOLID187, a 10-node tetrahedral structural solid.

The second element type, the SOLID186, is considered a higher version of the first one, and, in principle, they could offer more geometrical options than SOLID187 as this last one only can be tetrahedral and the other two can have the shape of a prism, a pyramid or the abovementioned tetrahedron.

In the case of the curved-cone shape, only the base and tip areas can be selected by the user, apart from the MN length or height. The curvature radius, as well as the curvature center, are fixed by the tool depending on the MN height (H) as five times H and half H (Z axis), respectively.

The first dimension that can be selected is the MN length or height, that has been limited between 100 μm and 1000 μm, which is an average range for microneedles intended for ISF sampling or drug delivery applications. In this case, the parameter can be manually established via a slider bar that has a default value of 400 μm, as can be seen in Figure 5. Moving down in the left side panel, the next two sliders permit the selection of the base and tip area of the MN. As the geometries included in the tool database are symmetrical in the X and Y axes, defining the area implies and automatic selection of the radius or side length.

The next slider bar permits choosing the meshing size, from 1 (not recommended for its inaccuracy) to 10 (longer computing time). Nevertheless, to deliver basic simulations and obtain preliminary results, a medium–low mesh size of 4 is selected by default.

As this tool is intended for solid and hollow MN design, there are two additional slider bars for selection of the hole or bore dimension and position, as shown in Figure 3. The default value for all the topologies is a null bore radius; i.e., a solid MN will be modeled if not specified otherwise. Definition of the limits for the slider that permits configuring the bore radius are linked with the value selected for the tip area in such a way that, in case of a centered hole (null offset), this will be circumscribed in the tip area. The bore offset parameter allows inclusion of a displacement in the hole center from the default origin of the coordinate system to a separate point in the X-axis, giving rise to an eccentric bore in the MN.

This design tool also offers the possibility of creating the model of an MN array by independently configuring the number of replicas in the X and Y axis and center-to-center interspacing, as shown in Figure 6. As indicated in the warning sentence above this part of the panel, the default values are “1”, indicating that a single MN will be included in the mechanical model for characterization unless defined otherwise.

There is a summary table below the spinners and text field for selection of the array options (Figure 7), where all the aforementioned modelling parameters are included. With the exemption of the MN shape, the rest of the table cells are editable, permitting manual insertion of the values via keyboard and an automatic update of the bar positions.

Once the whole structure of the MN has been configured, the “Generate Batch” button runs the MATLAB script that exports all the selected options to an ANSYS batch file including not only this dataset but also all the necessary instructions to run automatically both the static and linear buckling analyses. When the button is pressed, the path of the subsequent batch file appears underneath, as shown in Figure 8, so that the designer could check and execute separately its different modules if desired. Moreover, with the aim of facilitating and simplifying the simulation process, especially to non-expert users of ANSYS, if both the MATLAB app and ANSYS program are going to run on the same computer, the tool provides the command lines to directly copy (via the “Copy” button), paste and run the FEM simulation.

Figure 9 displays the von Mises stress distribution on the deformed shape of a conical hollow microneedle that has been designed automatically with the tool, considering an axial pressure of 3.18 Mpa applied to the tip area and a SOLID186 element. The dimensions of the MN correspond to the values previously seen in Figure 3, Figure 4, Figure 5, Figure 6 and Figure 7, i.e., height 600 μm, base area 31,416 μm^2^ (base diameter 200 μm), tip area 1257 μm^2^ (tip diameter 40 μm), mesh size 5, bore radius 10 μm, bore offset 40 μm.

It is worth mentioning that, in this first stage of development of the tool, a material selection option is not provided. Silicon is the manufacturing material considered for all the designs, in such a way that its Young’s modulus, yield stress, Poisson’s factor and density are values included by default in the batch file for simulation.

#### 3.1.2. Results Panel

In the right-side part of the main panel, there is an area only active once the ANSYS simulation has been finished. It is necessary to run both programs on the same computer so that the tool could read the results that, according to the generated batch, have been stored in a predefined folder.

Figure 10 shows the graph appearing in the upper zone of this panel, where the absolute Z stress along the X axis is represented, obtained from the static analysis with axial load. This figure permits rapid identification of the areas that are more significantly affected by the stress and if in any case the maximum yield stress of the material has been exceeded.

Below this Z stress graph, the panel includes another table (Figure 11) summarizing important information about the analysis. The first field corresponds to the computing time, the second one shows the element count, which obviously depends on the selected mesh size, and this one is followed by the values for the element warnings and errors reported during simulation. The data shown in Figure 11 correspond to the same simulation used as an example, performed by a desktop computer (Intel (R) Core (TM) i7-4790 CPU @ 3.60 GHz, RAM 8 GB).

Apart from data related to the simulation process, this table also includes results from the static and linear buckling analysis, such as the top element stress, the max Z stress and the X and Y displacements. The “Top Element Stress” is the maximum von Mises stress endured by one of the elements at the tip area, and it is retrieved directly from ANSYS once the static analysis is finished. The “Max Z stress” is calculated from the upper graph shown in Figure 10. The “X-displacement” and the “Y-displacement” values correspond to the tip of the MN, and both are obtained from linear buckling analysis.

## 4. Discussion

The graphical interface defined for the automated design tool permits a non-expert user to develop a mechanical model of MN arrays or single units that can be directly simulated in an FEM software with minimum effort and in a very visual way. Additionally, this tool reduces significantly the necessary time to analyze and compare results from simulation sets where a fundamental parameter that defines the geometry is changed to assess its impact on the mechanical strength of the structure, or for optimization proposes. If the designer wants to change a dimension of the structure, or even the geometry itself, but preserve the rest of parameters, there is no need of making significant changes to the design or even starting it again from zero.

From the review that has been presented in Section 2, it can be noted that many of the works that have been analyzed make use of CAD tools to create the MN geometrical models. Our proposal provides not only that feature but also the additional possibility of exporting and automatically simulating the structure of the MN, including material properties and meshing options, together with loading and boundary conditions. In addition, our MATLAB tool collects the simulation results so that analytical and/or experimental results could be easily compared, taking advantage of the high computing capacity of MATLAB software.

Another advantage of utilizing the proposed design tool is developing a dynamic simulation of the insertion process by combining the generated model for the microneedle or array with a previously created model of the skin. However, one of the fields of future work that have been foreseen is integrating two selecTable 3D models for the skin among the simulation options. One of the models would consider elastic response for the three layers of the skin, whereas the other one would be based on a more advanced configuration of hyperelastic behavior for the stratum corneum and dermis layers according to the existing proposals in the area. In both cases, an erosion algorithm would be advisable to better determine the insertion force, and to characterize the loading scheme the MN would be subjected to. In this sense, the fact that the tool can provide the model of an array besides a single MN would allow optimizing the insertion efficiency of the patch in the previous simulation, depending of the MN interspacing parameter, so as to avoid the aforementioned “bed of nails” effect.

This first version of the tool was focused on proving the feasibility of the proposal, and a limited number of key options were selected to be present in the main panels. In this way, a correct performance could be ensured before considering any future extension. For that reason, only those topologies that present symmetry in both the X and Y axis and a single height parameter were chosen for the current version. Therefore, all of them can be modeled on the basis of only three values, corresponding to the height and the tip and base areas. Among the most common topologies that can be found in the literature, the tapered-cone and the cylinder with beveled tip would be the next geometries selected to be included in the database of the tool. From the point of view of programming requirements, they are relatively easy structures to be modeled, but they would need additional configuration parameters, such as the height of the base cylinder and the tip angle, respectively. There are two options for that extension: expanding the current configuration panel to include additional slider bars or creating a third panel, only accessible when these geometrical options were selected.

Though later extended, this proposal was initially conceived to facilitate design of microneedles fabricated with MEMS-based manufacturing processes so that they could be integrated with a future microfluidic sensing device and an electronic processing system. That is the reason why silicon is the default material selected for modeling the MN in this first version of the tool. In any case, enhancing the versatility of the tool in a forthcoming version involves including other biocompatible materials to be selected by the designer. Moreover, an interesting feature to add in a future menu for material selection would be defining a new user’s materials with personalized values for Young’s modulus, Poisson’s factor, density or yield strength.

Regarding how to model hollow microneedles, there are several papers in the specialist literature proposing complex geometries for the inner bore, with the aim of improving sample extraction or drug delivery [11,35]. Our tool currently contemplates an approximation that could be easily achievable with CMOS compatible processes, or with additive manufacturing approaches. In this sense, future work could include an additional and exclusive panel for bore configuration that would consider modeling centered topologies with T- or Y-shaped single- or double-sided exits, and where parameters such as exit height or tilt angle (for Y-shaped case) could be adjusted.

Currently, the tool only considers axial load for the static and buckling analysis. Different methodologies have been proposed to characterize microneedles under bending conditions, and values and ways of application greatly differ among them. The authors consider that bending failure should be investigated due to possible incorrect application of the patch or even during its removal, especially for biosensing systems where the integrity of the sample is imperative. However, how to include this option in future versions of the tool is still undetermined, and further research is needed.

## 5. Conclusions

This paper presents a detailed review of the latest published proposals where an FEM-based analysis on microneedles has been included. The purpose of this 5-year review was establishing the minimum set of features needed for a software tool intended for automation of a common microneedle design process based on mechanical characterization.

This objective has been successfully achieved by developing a MATLAB design tool that permits definition of CAD models for commonly used microneedle geometries. Additionally, the proposal provides the additional feature of automated exportation of the MN structure and direct simulation by an FEM-based software of basic static and linear buckling analyses. In order for this simulation to be performed, the batch file generated by the tool for FEM analysis includes significant data, such as MN dimensions, material properties, meshing and loading options and boundary conditions.

The graphical interface developed for the tool allows non-expert users to conduct a mechanical characterization of MN arrays that are ready for simulation. Additionally, this software significantly reduces the necessary time and effort to perform topological changes for optimization purposes.

It is worth mentioning that this tool permits utilizing the generated model for the MN array with previously created skin models in order to implement a dynamic simulation of an insertion process.

This software was initially intended for design of MN arrays that were fabricated with CMOS-compatible technologies, and that is the reason why the default material currently included in its database is silicon. Future versions of the tool are planned to add other biocompatible materials, as well as new geometries, additional options for bore definition in hollow topologies, 3D skin models and bending analysis.

## Figures and Tables

**Figure 1 micromachines-14-00133-f001:**
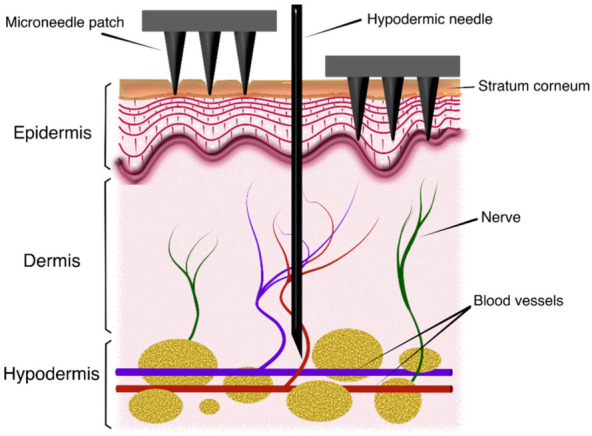
Schematic view of the insertion process of microneedle patches into skin.

**Figure 2 micromachines-14-00133-f002:**
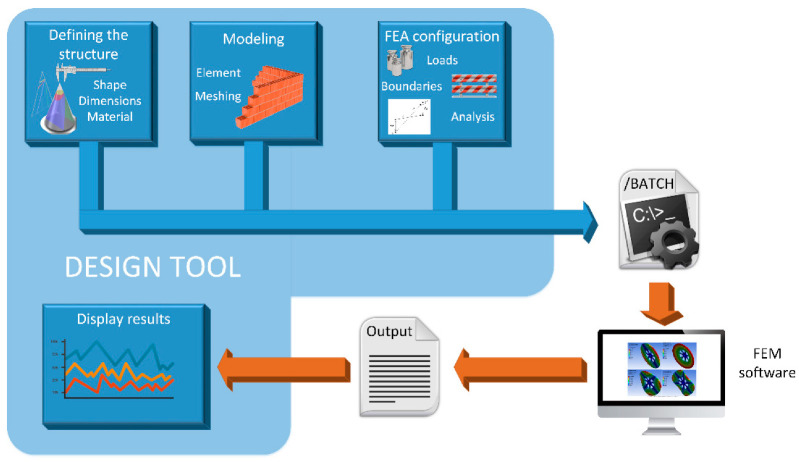
Workflow of the design process of microneedles.

**Figure 3 micromachines-14-00133-f003:**
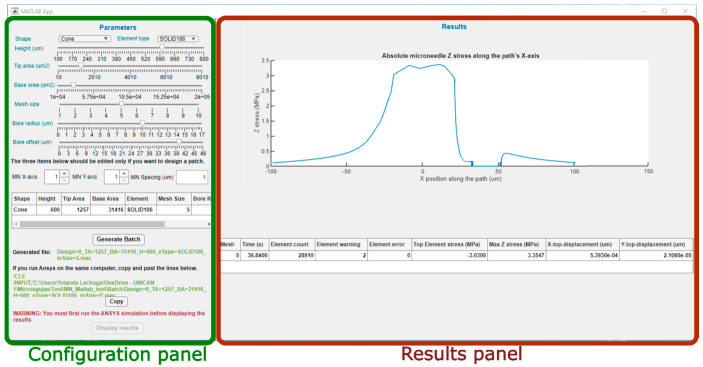
Main screen of the MATLAB tool and division in two panels.

**Figure 4 micromachines-14-00133-f004:**
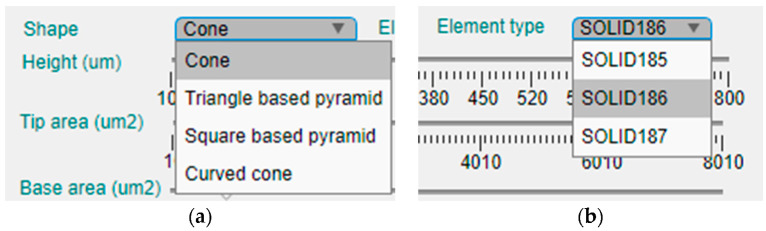
Enlargement of the upper part of the configuration panel and the two drop-down menus for selection of the: (**a**) shape of the MN; (**b**) element type for FEM simulation.

**Figure 5 micromachines-14-00133-f005:**
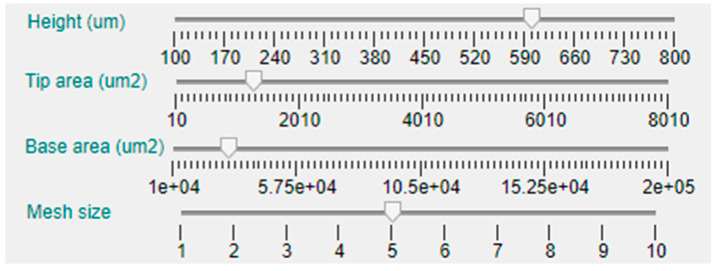
Details of the central area of the configuration panel.

**Figure 6 micromachines-14-00133-f006:**
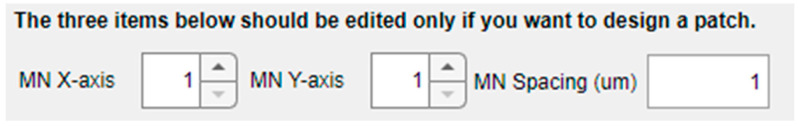
Options in the left panel for creating MN arrays.

**Figure 7 micromachines-14-00133-f007:**
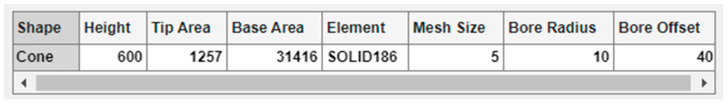
Enlargement of the summary table with its editable cells.

**Figure 8 micromachines-14-00133-f008:**
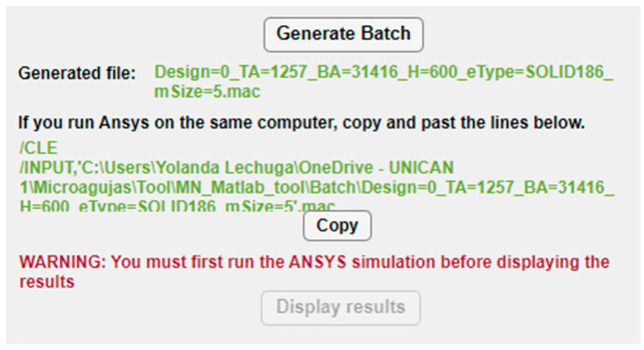
Buttons and instructions in the configuration panel for creation of the ANSYS batch file and simulation.

**Figure 9 micromachines-14-00133-f009:**
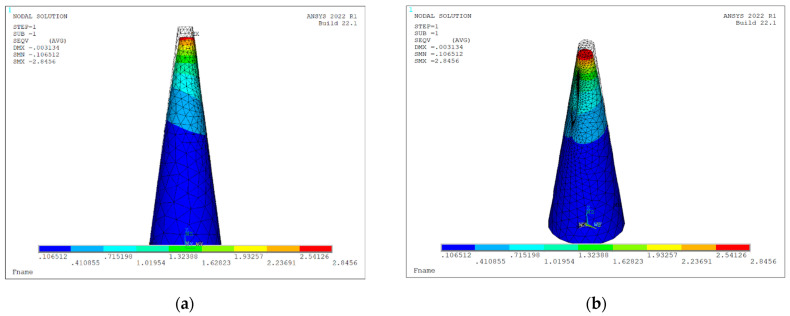
Von Mises stress distribution on deformed shape of a hollow conical MN (L = 600 μm, A_b_ = 31,416 μm^2^, A_t_ = 1257 μm^2^ R_h_ = 10 μm, H_os_ = 40 μm) under axial load of 3.18 mPa considering static and linear buckling analysis: (**a**) front view; (**b**) isometric view.

**Figure 10 micromachines-14-00133-f010:**
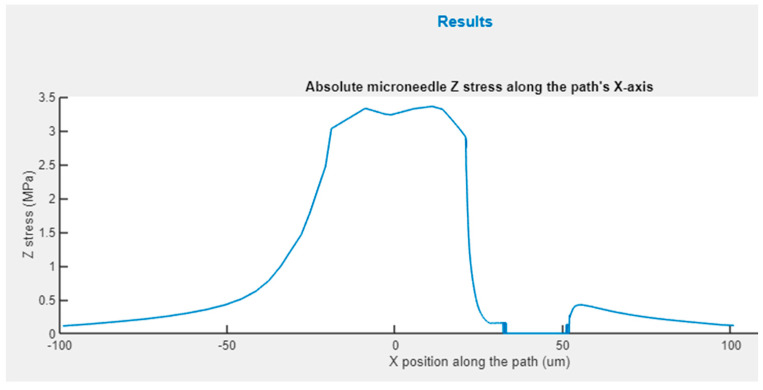
Z-stress distribution for an X-path cross-section of hollow conical MN.

**Figure 11 micromachines-14-00133-f011:**
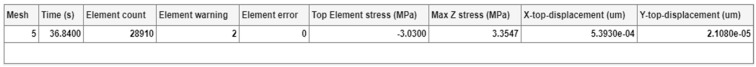
Details of the results table.

## Data Availability

The data that support the findings of this study are available from the corresponding author, YL, upon reasonable request.

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
