# Peer review of "Development of an Automated Design Tool for FEM-Based Characterization of Solid and Hollow Microneedles"

_micromachines, 2023, doi:10.3390/mi14010133_

Round 1

Author Response

Please, see the attachment. Thank you

Reviewer 2 Report

The authors have successfully explored the development of an automated design tool for FEM-based characterization of solid and hollow microneedles. Overall, this manuscript is very interesting and beneficial for developing the design of microneedles. This manuscript is very well and systematically written, also the English is very good. The authors have elaborated and explained the novelty of this manuscript comprehensively. However there are few punctuation errors, such as "of" in Line 109, the use of ":" Line 189, and the "et all" in Line 172 should be written "et al" as found in other citations. Moreover, a conclusion section should also be added to this manuscript. I recommend this manuscript to be published in Micromachines after completing the minor corrections mentioned above.

Author Response

Please, see the attachment. Thank you.
